# Retinal biological age correlates with bone mineral density and fracture risk score and predicts incident osteoporosis

Qingsheng Peng[1,2], Can Can Xue[1], Kenon Chua[2,6], Hengtong Li[4], Simon Nusinovici[1,5], Zhi Da Soh[1], Preeti Gupta[1], Ah Young Leem[7], Marco C.Y. Yu[1,5], Charumathi Sabanayagam[1,5], Tyler Hyungtaek Rim[5,7], Zhuoting Zhu[8], Ecosse Lamoureux[1,5], Ching-Yu Cheng[1,3,4,5]*

1 Singapore Eye Research Institute, Singapore National Eye Centre, Singapore, Singapore, 2 Clinical and Translational Sciences Program, Duke-NUS Medical School, Singapore, Singapore, 3 Centre for Innovation and Precision Eye Health, Yong Loo Lin School of Medicine, National University of Singapore, Singapore, Singapore, 4 Department of Ophthalmology, Yong Loo Lin School of Medicine, National University of Singapore, Singapore, Singapore, 5 Ophthalmology and Visual Sciences Academic Clinical Program (Eye ACP), Duke-NUS Medical School, Singapore, Singapore, 6 Department of Orthopedic Surgery, Singapore General Hospital, Singapore, Singapore, 7 Mediwhale Inc., Seoul, Republic of Korea, 8 Center for Vision Research Australia, Melbourn, Australia

* chingyu.cheng@nus.edu.sg

## Abstract

Osteoporosis often lacks accessible screening tools, leading to underdiagnosis and increased fracture risk. We explored the potential of a retinal aging biomarker, measured by the RetiAGE algorithm, in stratifying osteoporosis risk. Cross-sectional and prospective cohort study. The retinal biological aging biomarker, RetiAGE, indicates the probability of being older than 65 years, was derived from retinal photographs using a deep learning algorithm. In the cross-sectional PopulatION HEalth and Eye Disease PRofilE in Elderly Singaporeans (PIONEER) study with 1,965 participants with both retinal images and Dual-energy X-ray Absorptiometry (DEXA) measurements, we assessed the association of RetiAGE with bone mineral density (BMD) and BMD's standard deviation (SD) score (T-score), and major osteoporotic and hip fracture risk scores calculated from fracture assessment tool (FRAX) using linear regression models, and its association with osteoporosis using logistic model. In the prospective UK Biobank cohort with 43,938 participants with retinal photographs and without osteoporosis at baseline, we evaluated the association between RetiAGE and the onset of osteoporosis using multivariable Cox proportional hazard models. Subgroup analyses were performed by further adjusting for menopause, hormone replacement therapy and glucocorticoids in women. In the PIONEER study, older RetiAGE was inversely associated with BMD and T-scores in various femoral regions after adjusting for risk factors (all $p < 0.05$). Elevated RetiAGE was associated with an increased risk score of major osteoporotic and hip fractures ($\beta$ coefficients of 0.48 and 0.29, per SD increment, respectively). In the UK Biobank participants,

**Data availability statement:** The prospective data used in this study are available from the UK Biobank (https://www.ukbiobank.ac.uk/) under license. Access to UK Biobank data is open to bona fide researchers through an application process, subject to approval of a research proposal and agreement to the UK Biobank terms and conditions. The PIONEER is an ongoing population-based cohort study. In accordance with institutional and ethical requirements, individual-level data cannot be publicly released but can be made available to qualified researchers upon reasonable request (via ecosse.lamoureux@seri.com.sg) and completion of a data-sharing agreement with the Singapore Eye Research Institute (institution webpage: https://www.snec.com.sg/research-innovation; institution email: seri@seri.com.sg).

**Funding:** This work was supported by the National Medical Research Council of Singapore (NMRC/CIRG/1488/2018 to C.Y.C.). The funders had no role in study design, data collection and analysis, decision to publish, or preparation of the manuscript.

**Competing interests:** I have read the journal's policy and the authors of this manuscript have the following competing interests: T.H.R. was a former scientific advisor to Mediwhale and owns stock in the company. C.Y.C. has received consulting fees from Mediwhale and is a co-founder of Eye.AI. These relationships did not influence the study design, analysis, interpretation, or manuscript preparation. All other authors declare that no competing interests exist.

higher RetiAGE predicted future osteoporosis onset (hazard ratio, HR = 1.12, per SD increment, $p = 0.001$), with significant associations persisting in subgroup analyses ($p < 0.001$ in women; $p = 0.011$ in men). Accelerated retinal biological aging is associated with decreased BMD and an increased risk of osteoporosis and related fractures. Retinal age may provide a potential alternative for opportunistic risk screening.

## Author summary

Osteoporosis is a common condition that weakens bones and raises the risk of fractures, especially in older adults. However, many individuals are not diagnosed until after a fracture occurs, in part because the standard diagnostic test, Dual-energy X-ray Absorptiometry (DEXA), is not always readily accessible. We therefore investigated whether retinal photographs, taken from the back of the eye, could help identify people at higher risk of osteoporosis. This possibility arises from the idea that the retina may reflect the body's overall biological aging. Hence, we used an artificial intelligence–derived age marker, RetiAGE, to estimate retinal biological age and test the association between retinal age and osteoporosis. In the Singapore study of 1,965 older adults, older retinal biological age was associated with lower bone mineral density (BMD), lower BMD T-scores, and higher fracture risk scores. In the UK Biobank study of 43,938 participants, older retinal biological age also predicted a higher risk of developing osteoporosis over time, even after accounting for major risk factors. These findings suggest that retinal biological aging may reflect broader aging processes related to skeletal health. Retinal imaging may therefore provide a simple, non-invasive, and accessible way to support opportunistic screening for osteoporosis risk.

## Introduction

Osteoporosis is a systemic skeletal disease characterized by decreased bone mass and

   deterioration of bone tissue microarchitecture. [1] Globally, approximately 19.7% of the population is affected by osteoporosis, with the prevalence increasing with age. If left untreated, osteoporosis elevates the risk of major fractures, which can be life-threatening and result in considerable healthcare costs and loss of productivity. [2]

   Despite its high prevalence and serious consequences, osteoporosis often remains underdiagnosed. [3,4] This is because the gold-standard method for diagnosing osteoporosis-the measurement of bone mineral density (BMD) using Dual-energy X-ray Absorptiometry (DEXA) scans- is costly and not widely accessible. Therefore, it is typically recommended only for high-risk individuals, such as those with suspected fractures on X-rays or patients on long-term steroid therapy. [5] This

selective screening approach limits early detection in the general population, especially among the elderly who may not exhibit obvious risk factors. Consequently, many people are diagnosed with osteoporosis only after experiencing a fracture. Given its widespread prevalence and underdiagnosis, accessible and non-invasive screening modalities that can be broadly applied to identify at-risk individuals before fractures occur are in dire need.

Retinal images have been widely used to derived various systemic health information due to the close link between the eye and key systemic organs [6,7], the advances in artificial intelligence(AI), as well as the non-invasive, easily accessible nature of the fundus. [8] Recently, impaired retinal microvasculature has been identified as a risk factor for vertebral fractures in older adults. [9] Additionally, consistent retinal arterioles narrowing have been observed in osteoporotic females with primary hyperparathyroidism, irrespective of hypertension status. [10] Moreover, physical activity and outdoor sun exposure, key protective factors against osteoporosis, have been shown to influence retinal blood perfusion.[11,12] Together, these preliminary findings suggest that shared biological processes may link retinal health and skeletal integrity. However, direct evidence connecting retinal aging to bone mineral density loss remains limited, and few studies have examined whether a homogenous biological aging process underlies changes in both tissues.

In our previous work, we developed retinal biological age markers directly from retinal images using deep learning techniques. The retinal age gap and RetiAGE, derived from fundus photographs using Convolutional Neural Networks (CNNs), represent significant advancements in this field. [13,14] These markers have shown promise in predicting the risk of age-related diseases, including cardiovascular conditions [14,15], Parkinson's disease [16], and chronic kidney disease (CKD). [17] Furthermore, our genome-wide association studies (GWAS) identified variants in the IRF4 locus associated to retinal biological age. The IRF4 regulates the polarization of macrophages to osteoblast cells, which plays a pivotal role in osteoporosis development. [18,19] This connection positions retinal aging as a potential indicator for assessing low bone mineral density, offering a novel approach to identifying and managing osteoporotic risk.

Therefore, in this study, we aimed to investigate whether retinal biological aging, as represented by RetiAGE, was associated with bone mineral density and osteoporosis in a cross-sectional study (PopulatION HEalth and Eye Disease PRofilE in Elderly Singaporeans (PIONEER)) and whether an older retinal biological age predicted incident osteoporosis in the prospective UK Biobank cohort.

## Method

### Study design and ethics statement

We conducted the cross-sectional analysis in the population-based PIONEER study and the prospective cohort study in the UK Biobank.

The SingHealth Centralized Institutional Review Board approved the protocol of The PIONEER study, and North West Multi-centre Research Ethics Committee approved the UK Biobank study as a research tissue bank (Title of the database: UK Biobank: a large scale prospective epidemiological resource; REC reference: 21/NW/0157; IRAS project ID: 299116). Both studies were carried out following the tenets of the Declaration of Helsinki, and all participants provided written consent.

### Retinal biological aging marker (RetiAGE)

We previously created a deep learning model utilizing 129,236 retinal images from 40,480 Korean Health Screening study participants. The algorithm was designed to take retinal images from both eye into consideration to lower the possibility of missing out important aging signals. Based on macula-centered retinal photographs, this model was tasked to estimate the likelihood of an individual being 65 years or older. The 65 years as the threshold was set according to the definition of "Elderly Persons" by the Singapore Department of Statistics. [20] Deep learning algorithm for retinal biological age prediction used a previously developed deep learning model based on the Visual Geometry Group (VGG16) convolutional neural network architecture. Retinal photographs were resized to 300 × 300 pixels and underwent standard data augmentation,

including random cropping, rotation, flipping, and photometric adjustments. The network was trained from scratch using the Adam optimizer (learning rate $1 \times 10^{-4}$, batch size 16) and optimized using cross-entropy loss, with model selection based on validation performance. To enhance robustness across ethnicities and imaging conditions, local contrast normalization was applied by subtracting the local mean intensity. Model interpretability was explored using guided backpropagation with SmoothGrad. Full details of model architecture, training procedures, and internal validation have been reported previously. [14] RetiAGE was analyzed as a continuous probability score; therefore, the age threshold used during model training affects calibration and coefficient scaling but does not influence rank-based discrimination or association estimates.

## Cross-sectional analysis in the population-based PIONEER study

The PIONEER study is geographically-representative and population-based epidemiology study focusing on age-related disease and function decline in Asians aged over 60 years in Singapore. [21] The study has 3,152 participants from multi-ethnic (Chinese, Indian, Malay) background at baseline. The eye examination and DEXA scans were conducted on the same day for each participants at enrollment.

### Fundus photography and assessment of the retinal biological aging marker (RetiAGE)

After confirmed safe to dilate pupils by ophthalmologists, the participants' 45° field-of-view retinal photographs were taken after pupil dilation with the Canon CRDGi digital nonmydriatic retinal camera according to the Early Treatment for Diabetic Retinopathy Study (ETDRS) 7 field standard. In each eye, the macula-centered retinal photograph with the best quality judging by an in-house quality control model with a threshold of 0.593, were chosen to generate the RetiAGE scores. The in-house quality control model was trained on over 20,000 quality controlled retinal images with a Swin transformer. The model achieved an AUC of 0.97 classifying the retinal image readability. The RetiAGE scores for individual participants were calculated as the average of two eyes (between two eyes image, intraclass correlation coefficient, ICC = 0.65, 95% confidence interval 0.62-0.68; mean difference = 0.04 ± 0.23).

### Dual-energy X-ray Absorptiometry scans and bone mineral density

The DEXA scans (Hologic Discovery-W; Hologic Inc, Bedford-MA) were taken with X-rays (~0.005 to 0.01 millisievert, Horizon User Guide MAN-08072–002 Rev 003) for the PIONEER participants. [21] The bone mineral content (BMC), bone area, and bone mineral density (BMD) were measured from the DEXA scans. The BMD T-score represents the number of standard deviations by which an individual's measured BMD differs from the mean BMD of a young-adult reference population, as defined by the manufacturer-provided Third National Health and Nutrition Examination Survey (NHANES III) norms. DEXA outputs were screened for implausible or non-numeric values indicative of data entry or extraction errors (e.g., string entries in numeric fields). Values consistent with physiological ranges were retained.

### Definition of osteoporosis and the calculation of 10-year probability of MOF and HF

The diagnosis of osteoporosis for the PIONEER study was based on the World Health Organization (WHO) and the International Osteoporosis Foundation (IOF) criteria of whether BMD of the participant's femoral neck reached -2.5 SD or lower. [22] Diseases such as rheumatoid arthritis and primary hyperparathyroidism can also impact BMD and develop osteoporosis. [23] Thus, the secondary osteoporosis was excluded from this study. The major osteoporotic fracture (MOF) and the hip fracture (HF) risk score of each participant was measured by a radiologist on-site using the Fracture Risk Assessment Tool (FRAX, Centre for Metabolic Bone Diseases, University of Sheffield, UK). [24] The calculator, derived from country-specific fracture and mortality data to estimate the 10-year probability of MOF and HF, includes age, sex, weight, height, fracture history, smoking, alcohol drinking, glucocorticoids usage, rheumatoid arthritis, secondary osteoporosis diagnosis and femoral neck BMD. The FRAX were applied by an radiologist during the DEXA examination on each participant.

 

## Inclusion of other covariates

Previous studies showed that old age, female gender, ethnicity, family history, low calcium intake, vitamin D deficiency, low physical activity, smoking, excessive alcohol consumption (more than two glasses of 120 ml of wine per day), diabetes mellitus (DM), hypertension, smaller body size, menopause, hormone deficiency, the use of glucocorticoids and similar medicine lead to increased risk of osteoporosis. [25] In the PIONEER study, risk factors for osteoporosis are limited to data from measurements during research enrollment and self-reported questionnaires. Hence, only risk factors from PIONEER that align with the referred definitions were included in the analysis. All included female participants passed the age of menopause. Risk factors for low BMD and osteoporosis included calendar age, gender, ethnicity, calcium intake (estimated mg/day), light and moderate physical activity (measured by hours of activity in a typical week), smoking, DM, hypertension, body mass index (BMI), and glucocorticoids and similar medicines usage (participants reported glucocorticoid usage and a binary answer of yes or no was given) were extracted for the analysis.

## Exclusion criteria

In the PIONEER study, we performed quality control over the retinal photos. Poor-quality retinal images that were deemed ungradable by an in-house image quality grading deep-learning model with a threshold of 0.593 were excluded. The participants with readable retinal photos from only one eye were excluded. We excluded participants who did not go through DEXA examination. The participants with secondary osteoporosis upon DEXA examination were also excluded from the analysis. The details are shown in S1 Fig.

## Statistical analysis

Analyses were conducted using R Programming (Version 4.3.2). Normally distributed continuous variables were shown as mean and standard deviation (SD) and non-normally distributed variables were shown as median and interquartile range (IQR). Categorical variables were shown as frequency and corresponding percentages.

In the PIONEER study, we first transformed the RetiAGE probability score to standard deviation scores (z-scores). Then, we evaluate the association between RetiAGE z-scores and DEXA parameters (such as BMD, T-scores and BMC across different areas in the DEXA scan, including lumbar vertebra L1 to L4, pelvis, right and left leg, and neck, inter, Ward's and troch area of the hip) using General Linear Model (GLM) and adjusted for multiple testing with False Discovery Rate method (FDR). We further adjusted for risk factors, including calendar age, gender, calcium intake, light and moderate physical activity, smoking, DM, hypertension, BMI, and glucocorticoids and similar medicine usage. We compared the difference in RetiAGE z-scores between the participants with osteoporosis and those without using GLM model, adjusted for osteoporosis risk factors. Further associations between the MOF and HF risk scores and RetiAGE z-scores were done to assess the added value of retinal biological age in assessing the likelihood of major osteoporotic fracture and hip fracture.

## Prospective cohort analysis in UK Biobank

The UK Biobank database covers over 500,000 residents aged 40–70 years. Participants at the baseline had their retinal photograph taken were included. The characteristics at the baseline and longitudinal data from hospital inpatient records and primary health care units in the UK were included in this study. [26]

## Fundus photography and assessment of the retinal biological aging marker (RetiAGE)

The Non-mydriatic, 45° field-of-view colour fundus photos centered on the macula were taken from 68,083 participants at the enrollment using the Topcon 3D OCT-1000 Mk2 device. The retinal photos were used to generate the eye-level RetiAGE score. The average score of the two eyes from each participant was calculated as the RetiAGE score for further analysis.

### Definition of incident osteoporosis

For the UK Biobank study subjects, data on hospital admissions were accessible until March 18, 2020, when the analysis was conducted. The diagnosis of osteoporosis in the UK Biobank study was based on the ICD 10 codes from the inpatient data (data field ID: 41270 and 41271). The osteoporotic fracture events were generated using the same definitions from previous study based on ICD 10 codes. [27] The follow-up time was calculated as the period between diagnosis recording day and the enrollment day for the uncensored participants and the period between end of observation and the enrollment day for the right censored participants.

### Inclusion of other covariates

Included risk factors of osteoporosis from UK Biobank were calendar age, gender, BMI, history of diabetes and hypertension, smoking status, light and moderate physical activity (the metabolic equivalent of task, MET) and under the treatment of glucocorticoids and similar medicine, which include prednisone, methylprednisolone, triamcinolone, dexamethasone, budesonide, prednisolone, hydrocortisone, betamethasone, cortisone, deflazacort. Calcium intake from "estimated food nutrients yesterday" only has 70,680 out of over 500,000 participants' data at the baseline. Thus, calcium intake was excluded from the analysis. For sub-group analysis of the participants of the female gender, we further included hormone-replacement therapy (HRT) and menopause (All data-field ID and coding from UK Biobank can be found in S1 Table). We further generated the Osteoporosis Self-assessment Tool (OST) scores from calendar age, gender and weight using published formular to test whether RetiAGE can assist OST on osteoporotic outcomes. [28]

### Exclusion criteria

In the UK Biobank cohort, we excluded the 337 participants whom were diagnosed with osteoporosis before enrollment. We performed the quality control process using the same in-house model with the same quality threshold of 0.593. Participants with missing values for the risk factors or poor-quality retinal photos were also excluded (S1 Fig).

### Statistical analysis

In the UK Biobank cohort, we used Cox proportional hazard models to stratify future risk of osteoporosis by RetiAGE quartiles, adjusting for potential risk, including calendar age, gender, BMI, history of diabetes and hypertension, smoking status, light and moderate activity MET and glucocorticoids and similar medicine. The trend of RetiAGE quartiles on osteoporosis risk was tested with quartile set to continuous variable. We also tested the performance of RetiAGE z-score on stratifying the risk of osteoporosis with Cox PH model, after adjusting for risk factors. To visualize effect sizes across predictors measured on different scales, we additionally calculated the hazard ratio (HR) per 1 standard deviation SD increase in each predictor. Continuous covariates were standardized using their empirical SDs. Binary covariates were coded as 0/1 indicators, with SD defined as $\sqrt{(p(1-p))}$, where $p$ represents the prevalence of the binary risk factor; this rescales effects onto a common SD unit without altering the fitted Cox model. Network reclassification indexes (NRI) at 10 years were estimated by comparing the Cox proportional hazard (PH) model with RetiAGE to the model without. We further tested whether RetiAGE improves the predictive performance of the OST risk categories on predicting incident osteoporotic fractures using the Cox PH models. Insufficiency in estrogen/progestogen and under glucocorticoids treatments were both medical conditions that can lead to osteoporosis [29], and potential interaction between the HRT and glucocorticoids exist [30]. Hence, we conducted sensitivity analyses in the women to further adjust for the treatment of glucocorticoids and similar medicines, HRT, and menopause. In men, only the treatment of glucocorticoids and similar medicines was further adjusted for in the analysis. We also conduct sensitivity analyses which only included the White population (defined as British, Irish, White, or any other white background, S5 Table), participants without ocular conditions potentially affecting retinal appearance (S6 Table and S7 Table) and participants who aged 60 and older (S8 Table, variable coding in S1 Table).

For the GWAS analysis, we included 45,496 White participants in the UK Biobank with readable retinal photos from both eyes. We first calculated the heritability for RetiAGE score. The single nucleotide polymorphism (SNPs) with an imputation quality score R2 below 0.3 and a minor allele frequency (MAF) less than 1% were excluded from the analyses. A linear-mixed model used applied to account for relatedness, as represented by the genetic relationship matrix. The analyses were then conducted using BOLT-LMM software under an infinitesimal mixed model and adjusted for sex, age, genotype array, and up to 20 principal components to account for population substructures within the European population. [31]

## Results

In PIONEER, 465 (23.7%) of the 1,965 participants whose mean age of 72.5 +/- 8.2 had osteoporosis. Of the 1,754 participants from PIONEER aged 65 and over, 420 (23.9%) were diagnosed with osteoporosis from DEXA. In the UK Biobank study, out of 43,938 participants aged 56.6 +/- 8.1 without osteoporosis at the baseline, 1,492 (3.4%) participants developed osteoporosis during the average 12.2 +/- 1.8 years of follow-up. The baseline demographics of the two studies are presented in **Table 1**. The RetiAGE distribution among the White, and Singaporean Chinese, Indian, and Malay were shown in the S4 Fig.

### Cross-sectional analysis on BMD in PIONEER study

In the linear regression model, the BMD and T-scores of the Ward's, troch, neck and inter regions of the femur, hip, and pelvis were all negatively associated with RetiAGE after adjusting for multiple testing using the FDR method (all $p < 0.001$, S2 Fig and S2 Table). The associations of BMD and T-score of the troch, inter and Ward's regions of femur and hip bone

**Table 1. Demographics and clinical characteristics of the study populations.**

|  | PIONEER | UK Biobank |
| --- | --- | --- |
| Osteoporosis |  |  |
| At baseline | 465 (23.7) | – |
| In the follow-up | – | 1,492 (3.4) |
| Women | 1,053 (53.6) | 19,999 (45.5) |
| Diabetes | 559 (28.5) | 2,040 (4.6) |
| Hypertension | 1,075 (54.7) | 11,079(25.2) |
| Current smoking | 171 (9.0) | 4,140 (9.4) |
| Glucocorticoids | 51 (2.2) | 815 (1.9) |
| Calendar age, years | 72.5 (8.2) | 56.2 (8.1) |
| Calcium intake, mg/day | 554.4 (374.6 – 689.8) | – |
| Light activity, hrs/week | 23.5 (14.0 – 37.0) | – |
| Moderate& vigorous activity, hrs/week | 0.0 (0.0 – 4.0) | – |
| MET, mins/week |  |  |
| Walking | – | 639.0 (330.0 – 1386.0) |
| Moderate | – | 480.5 (160.0 – 1200.0) |
| RetiAGE score | 0.3 (0.1 – 0.5) | 0.3 (0.0 – 0.6) |

PIONEER, PopulatION HEalth and Eye Disease PRofilE in Elderly Singaporeans study. MET, metabolic equivalent of task. Light, moderate and vigorous activity in PIONEER was measured as total hours in a typical week. RetiAGE score is the algorithm-generated probability score ranging from 0 to 1.

Categorical variables were shown as frequency and corresponding percentages. Normally distributed continuous variables were shown as mean and standard deviation (SD) and abnormally distributed variables were shown as median and interquartile range (IQR).

aOsteoporosis was defined using the WHO guideline.

with retinal age stayed significant after adjusting for osteoporotic risk factors ($\beta \times 10^{-2}$ for T-score per SD increment of Reti-AGE ranged from -10.96 to -6.82; adjusted $p$ ranged from 0.022 to 0.034, **Table 2**).

### Association between RetiAGE and odds of osteoporosis and fracture

After adjusting for age, gender, and other risk factors, the RetiAGE scores were positively associated with the diagnosis of osteoporosis (odds ratio, OR= 2.13, 95% confidence interval, CI, 1.12 to 4.05). The MOF and HF risk scores were also positively correlated to RetiAGE scores (**Table 3**).

### Longitudinal prediction of osteoporosis and fracture risk in the UK Biobank

In the Cox PH model, for every SD increase in RetiAGE, we observed a significant 12% higher risk of osteoporosis after adjusting for age and gender (S2 Table). The associations remained significant after further adjusting for BMI, history of diabetes and hypertension, smoking status, and light and moderate physical activity (HR per SD = 1.12, 95% CI range from 1.05 to 1.20). When RetiAGE was analyzed in quartiles, a significant dose–response relationship was observed, with progressively higher risks from the lowest to the highest quartile (p for trend < 0.001). Participants in the upper RetiAGE quartiles (quartiles 3 and 4) had significantly higher osteoporosis risk compared with those in the lowest quartile after multivariable adjustment (**Table 4**). In multivariable models, established demographic and anthropometric factors—particularly age (HR per SD = 2.23), gender (HR per SD = 2.37), and body mass index (HR per SD = 0.69)—remained the dominant predictors of osteoporosis risk. Lifestyle factors such as current smoking (HR per SD = 1.07) and lower physical activity (HR per SD = 0.94) were also associated with increased risk, but showed weaker associations compared with demographic factors and retinal biological age. The Cox PH model with RetiAGE quartiles showed significantly better classification performance than the model without at 10 years (Overall NRI = 2.5%, S3 Fig).

To further evaluate whether retinal biological age provides incremental prognostic information beyond existing screening tools, we examined the performance of the Osteoporosis Self-assessment Tool (OST) with and without the inclusion of RetiAGE (S2 Table). In models without RetiAGE, higher OST risk levels were associated with a substantially increased risk of osteoporotic fracture (HR = 2.40; 95% CI, 2.14–2.69), with a concordance index (C-index) of 0.585. When Reti-AGE was added to the model, OST remained significantly associated with fracture risk (HR = 2.09; 95% CI, 1.86–2.35), and RetiAGE independently contributed to risk prediction (HR per SD = 1.32; 95% CI, 1.25–1.40). The combined model

**Table 2. Association between RetiAGE z-scores and BMD in multi-variable models.**

| DEXA parameters | BMD | | | T-score | | |
|---|---|---|---|---|---|---|
| | β coefficients [a] | Std.error[a] | p | β coefficients [a] | Std.error[a] | p |
| Neck | -7.56 | 4.07 | 0.064 | -6.94 | 3.46 | 0.045 [c] |
| troch | -8.49 | 3.82 | 0.026 [c] | -8.43 | 3.67 | 0.022 [c] |
| Inter | -10.96 | 5.35 | 0.040 [c] | -7.80 | 3.73 | 0.036 [c] |
| Ward's | -9.56 | 4.83 | 0.048 [c] | -7.30 | 3.45 | 0.034 [c] |
| Total femur | -9.19 | 4.44 | 0.039 [c] | -7.73 | 3.64 | 0.034 [c] |
| Pelvis [b] | -6.82 | 5.60 | 0.224 | – | – | – |

DEXA, dual-energy X-ray absorptiometry; BMD, bone mineral density, mg/cm²; T-score, standardized score for BMD. Std.error, standard error. β coefficients indicates the changed amount of the DEXA parameters with each standard deviation increment of RetiAGE.

The model adjusted for age, gender, Calcium intake (mg/d), diabetes, hypertension, smoking status, light, and moderate activity (hours/week) and glucocorticoids usage. *P* was adjusted for multiple testing using FDR method.

[a]Parameters have been multiplied by $10^{-2}$ for readability

[b]The pelvis bone T-score was not provided by the DEXA in PIONEER study.

[c]Statistically significant difference at *p* < 0.05.

**Table 3. Retinal age is associated with osteoporosis and risk score of osteoporotic fracture after adjusting for risk factors in the cross-sectional PIONEER study.**

| | Osteoporosis [a] | | | Major osteoporotic fracture (%) | | | | Hip fracture (%) | | |
|---|---|---|---|---|---|---|---|---|---|---|
| | OR | 95%CI | p | β coefficients | Std.error | p | β coefficients | Std.error | p |
| RetiAGE [b] | 1.19 | 1.00-1.42 | 0.021[d] | 0.48 | 0.18 | 0.007 [d] | 0.29 | 0.12 | 0.014 [c] |
| Age, year | 1.02 | 1.00-1.04 | 0.100 | 0.18 | 0.02 | <0.001[d] | 0.13 | 0.02 | <0.001[c] |
| Gender (women) | 3.03 | 0.23-0.47 | <0.001[d] | 5.02 | 0.32 | <0.001[d] | 1.47 | 0.22 | <0.001[c] |
| Weight, kg | 0.92 | 0.91-0.94 | <0.001[d] | -0.07 | 0.01 | <0.001[d] | -0.05 | 0.01 | <0.001[c] |
| Calcium intake, mg/day | 1.00 | 1.00-1.00 | 0.169 | 0.00 | 0.00 | 0.454 | 0.00 | 0.00 | 0.315 |
| Diabetes | 0.82 | 0.58-1.15 | 0.262 | -0.73 | 0.31 | 0.008 [d] | -0.48 | 0.20 | 0.019 [c] |
| Hypertension | 1.14 | 0.85-1.53 | 0.598 | -0.10 | 0.29 | 0.182 | -0.20 | 0.19 | 0.281 |
| Currently smoking | 0.88 | 0.61-1.27 | 0.566 | 0.05 | 0.37 | 0.625 | 0.20 | 0.24 | 0.396 |
| Light activity, hrs/week | 0.99 | 0.98-1.00 | 0.128 | 0.00 | 0.01 | 0.979 | 0.00[e] | 0.01 | 0.687 |
| Moderate activity, hrs/week | 0.99 | 0.97-1.01 | 0.215 | 0.00[e] | 0.02 | 0.942 | 0.00 | 0.01 | 0.698 |
| Glucocorticoids | 0.53 | 0.19-1.50 | 0.234 | 2.99 | 0.95 | 0.002 [d] | 0.93 | 0.63 | 0.140 |

OR, odds ratio. 95% CI, 95% confidence interval. Std.error, standard error.

Age, gender, weight, calcium intake, diabetes, hypertension, smoking status, light and moderate activity(hours/week), and glucocorticoids were adjusted in the analysis.

Osteoporosis is a binary diagnosis. Hence, OR and 95% CIs are presented. Major osteoporotic fracture and hip fracture are risk scores shown in percentages that generated from the FRAX tool. Hence, β coefficients and standard errors are presented.

[a]Osteoporosis was defined according to WHO guideline of femur neck T-score less than -2.5.

[b]RetiAGE was transformed into standardized z-scores, varying from -3 to +3.

[c]Statistically significant difference at $p<0.05$.

[d]Estimate is negative but rounds up to 0.00

demonstrated improved discrimination, with the C-index increasing from 0.585 to 0.635, indicating better risk stratification when retinal biological age was considered alongside OST.

## Sensitivity analysis in women's and men's subgroups

In the women's sub-group, after further adjusting for history of menopause, history of HRT usage, and treatment of glucocorticoids and similar medicines, the RetiAGE scores were still associated with elevated risk of osteoporosis. With each SD increase in RetiAGE, the risk of osteoporosis increases by 10%. In the men's sub-group, after further adjusting for glucocorticoids and similar medicine, each SD increase in RetiAGE was associated with a 25% increased risk of osteoporosis. In both sub-groups, glucocorticoid usage was significantly associated with osteoporotic risk (**Table 5**).

## Sensitivity analyses excluding ocular conditions potentially affecting retinal appearance

To evaluate whether the observed associations were driven by ocular conditions that could influence retinal image characteristics, we conducted sensitivity analyses in participants without major eye diseases and in participants without a history of cataract surgery in the UK Biobank (S6 Table and S7 Table). In both restricted cohorts, RetiAGE remained significantly associated with incident osteoporosis after multivariable adjustment. Among participants with healthy eyes, each standard deviation increase in RetiAGE was associated with a higher risk of osteoporosis ([HR = 1.12; 95% CI, 1.03–1.21; $p=0.009$). Similarly, in participants without cataract surgery, RetiAGE showed a comparable association with osteoporosis risk (HR = 1.14; 95% CI, 1.06–1.22; $p<0.001$). We also applied cross-sectional sensitivity analysis in participants without glaucoma, age-related macular degeneration, and diabetic retinopathy in the PIONEER cohort (RetiAGE & osteoporosis, OR=1.21; 95% CI, 1.01–1.46; $p=0.367$, S4 Table). The effect estimates for other covariates, including age, gender, and

**Table 4. RetiAGE predicts the risk of osteoporosis after adjusting for risk factors in the prospective UK Biobank cohort.**

| | HR | 95% CI | p |
|---|---|---|---|
| RetiAGE (continuous) [a] | 1.12 | 1.05-1.20 | <0.001[b] |
| RetiAGE(quartile) [c] | | | |
| Quartile 1 | 1.00 | [Reference] | – |
| Quartile 2 | 1.20 | 0.98-1.47 | 0.081 |
| Quartile 3 | 1.26 | 1.02-1.56 | 0.031 [b] |
| Quartile 4 | 1.40 | 1.12-1.75 | 0.003 [b] |
| Age, year | 1.11 | 1.09-1.11 | <0.001[b] |
| Gender (women) | 5.61 | 4.74-6.64 | <0.001 [b] |
| BMI, kg/m$^2$ | 0.93 | 0.91-0.94 | <0.001 [b] |
| DM history | 1.22 | 0.90-1.65 | 0.166 |
| HTN history | 1.11 | 0.97-1.28 | 0.101 |
| Current smoking | 1.16 | 1.03-1.31 | 0.014 [b] |
| MET (moderate), mins/week | 1.00 | 0.99-1.00 | 0.866 |
| MET (walking), mins/week | 0.99 | 0.99-1.00 | 0.008 [b] |

HR, hazard ratio; 95% CI, confidence interval; BMI, body-mass index; DM, diabetes mellitus; HTN, hypertension; MET, metabolic equivalent, measured in minutes.

Age, gender, BMI, diabetes, hypertension, smoking status, and METs of walking and moderate activity(minutes/week) were adjusted in the analysis.

C-index = 0.787, standard error = 0.006.

[a]RetiAGE was transformed into standardized z-scores, varying from -3 to +3.

[b]Statistically significant difference at p < 0.05.

[c]p for trend is < 0.001.

**Table 5. Sub-group analysis on risk of future osteoporosis in men and women's population of UK Biobank.**

| | Women | | | Men | | |
|---|---|---|---|---|---|---|
| | HR | 95% CI | p | HR | 95% CI | p |
| RetiAGE [a] | 1.10 | 1.02-1.18 | <0.001 [b] | 1.25 | 1.05-1.49 | 0.011 [b] |
| Age, year | 1.11 | 1.10-1.12 | <0.001 [b] | 1.05 | 1.02-1.08 | <0.001[b] |
| BMI, kg/m$^2$ | 0.92 | 0.91-0.94 | <0.001 [b] | 0.92 | 0.88-0.96 | <0.001[b] |
| DM history | 1.09 | 0.76-1.56 | 0.635 | 1.24 | 0.70-2.19 | 0.458 |
| HTN history | 1.07 | 0.92-1.24 | 0.377 | 1.42 | 1.01-1.98 | 0.042 [b] |
| Current smoking | 1.13 | 0.99-1.27 | 0.053 | 1.33 | 0.94-1.87 | 0.110 |
| MET (moderate), mins/week | 1.00 | 1.00-1.00 | 0.143 | 1.00 | 1.00-1.00 | 0.499 |
| MET (walking), mins/week | 1.00 | 1.00-1.00 | 0.144 | 1.00 | 1.00-1.00 | 0.575 |
| Had menopause | 1.09 | 0.97-1.23 | 0.129 | – | – | – |
| HRT treatment | 1.07 | 0.94-1.21 | 0.304 | – | – | – |
| Glucocorticoids & similar medicine | 2.35 | 1.65-3.34 | <0.001[b] | 6.80 | 3.84-12.04 | <0.001[b] |

HR, hazard ratio; 95% CI, confidence interval; BMI, body mass index; DM, diabetes mellitus; HTN, hypertension; MET, metabolic equivalent, measured in minutes; HRT, hormone replacement therapy.

Age, gender, BMI, diabetes, hypertension, smoking status, and METs of walking, moderate activity(minutes/week) and glucocorticoids usage were adjusted in the analysis, HRT were further adjusted in the women subgroup.

[a]RetiAGE was transformed into standardized z-scores, varying from -3 to +3.

[b]Statistically significant difference at p < 0.05.

other risk factors, were consistent with those observed in the *p*rimary analysis, suggesting that the association between retinal biological aging and osteoporosis risk was robust to exclusion of participants with ocular conditions that may alter retinal appearance.

### Sensitivity analysis in participants aged over 60 years

Given that osteoporosis predominantly affects older adults, we further performed a sensitivity analysis restricted to participants aged over 60 years at baseline (S8 Table). In this older subgroup, RetiAGE remained independently associated with incident osteoporosis (HR = 1.12; 95% CI, 1.03–1.21; *p* = 0.005) after adjustment for demographic, metabolic, and lifestyle factors. Age, female gender, and lower BMI continued to show strong associations with osteoporosis risk in the age-restricted cohort. These findings indicate that the association between accelerated retinal biological aging and osteoporosis risk persists in older adults, supporting the consistency of the observed relationship across age strata.

### Genome-wide association analysis of RetiAGE

Genome-wide association analysis identified multiple loci associated with the RetiAGE score (S9 Table). Lead variants were observed across several chromosomes and were predominantly located in intronic or intergenic regions. SNP-level association statistics, including effect allele frequencies, effect size estimates (*β*), standard errors, and association *p*-values derived from BOLT-LMM, are reported in S9 Table. Several lead variants mapped to or were located near genes such as IRF4, CTNNB1, HERC2, and SH3YL1. The SNP-based heritability of RetiAGE was estimated to be $h^2 = 0.137$ (standard error, SE = 0.014), indicating a measurable genetic contribution to inter-individual variation in retinal biological age.

## Discussion

This study aims to reveal the association between retinal aging and bone health, including osteoporosis. We presented the cross-sectional connection between retinal aging and bone health by analyzing RetiAGE and Bone Mineral Density (BMD) across multiple bone areas. The association between RetiAGE and T-scores, representing the standard deviation of BMD compared to the average BMD in the same bone area of healthy thirty-year-old adults, was also significant. Retinal age showed added value in evaluating the risk of major osteoporotic bone fractures and hip fractures, as represented by the MOF and HF risk scores calculated from the FRAX tool. We then demonstrated the predictive power of the retinal aging marker for the future onset of osteoporosis. Our investigation also covered whether the elevated risk from higher retinal aging speed varies between men and women, considering that women included in the UK Biobank were mainly around the age of menopause. The results indicated that the association persists after adjusting for common osteoporosis risk factors and female-specific risks, such as menopause and Hormone Replacement Therapy (HRT).

The BMD in bones susceptible to osteoporosis gradually declines with age. [32] Lower BMD leads to a higher risk of osteoporosis, endangering the elderly with an increased chance of bone fracture and worse outcomes afterwards. [33] However, using DEXA to screen and diagnose osteoporosis and osteopenia is expensive, involves radiation and requires radiologists on-site for the examination. [5,34] On the other hand, indicators of biological aging like telomere length and DNA methylation prove to accelerate the speed of BMD loss [35,36]. Multiple systemic diseases such as cardiovascular disease [14], Parkinson's disease [16], stroke [15] and chronic kidney diseases [17] were reported to be associated with faster retinal biological aging. RetiAGE scores, which can be easily extracted from standard retinal photos, offer a non-invasive, low-cost, repeatable measure. Our study results highlight the potential of using retinal photos to screen for pre-clinical osteoporotic patients.

The retina is prone to environmental damage and serves as a "window" to show accumulated effects. For example, retinal blood flow measured by optical coherence tomography angiography (OCTA), changes rapidly after exercises. [11] Better retinal vasculature was found in those who exercise at higher intensity [37]. Apart from physical exercise, outdoor sun exposure is critical in retina development and eye health. Adequate duration of outdoor activities can effectively reduce

the occurrence and progression of myopia by alleviating eye muscle fatigue, relaxing accommodation, increasing choroidal blood flow, focal dopamine release, and stimulating retinal pigment epithelial cells [38], while extensive outdoor sun exposure, on the contrary, increases the risk of age-related macular degeneration in elderlies. [12] Coincidentally, lack of physical exercise and sun exposure are major risk factors for low BMD. Regular physical exercise can help increase BMD and prevent osteoporosis in people at risk [39], and sun exposure provides a way to supplement vitamin D in addition to diet. Other risk factors for low BMD, such as smoking and metabolic diseases, also have accumulative effects on the retina. [40,41] Although these lifestyle risks are considered modifiable, they are difficult to compile and quantify, making retinal biological age a potential surrogate marker.

In addition to environmental factors, retinal biological age shares common genetic risks with bone health. For instance, our GWAS results revealed the IRF4 as one of the genes connected to accelerated eye aging. The IRF4 gene regulates the microglia activation on the retina as well as the osteoblast and osteoclast differentiation. [42,43] Microglia activation plays a major role in retinal degeneration [44] while dysregulated osteoblast and osteoclast activity contributes to bone remodeling imbalance and osteoporosis. Beyond IRF4-related immune regulation, several systemic aging pathways may plausibly link retinal and skeletal health. Chronic low-grade inflammation, microvascular dysfunction, and oxidative stress are well-recognized contributors to both retinal degeneration and bone mineral loss. [45,46] These processes can impair retinal perfusion, disrupt neurovascular integrity, and accelerate photoreceptor and microglial dysfunction, while concurrently promoting bone resorption and reducing bone formation. Consistent with this concept, epidemiological studies have reported increased osteoporosis risk among patients with dry eye disease [47] and a higher prevalence of age-related macular degeneration among postmenopausal women with osteoporosis, [48] with impaired immune and inflammatory regulation proposed as shared mechanisms. Supporting a broader biological overlap, gene enrichment analyses from prior GWAS of retinal aging markers have also identified associations with non-ocular pathways, including bone mineralization and systemic metabolic processes. [18,19] Collectively, these findings suggest that retinal biological aging may reflect multisystem aging processes that influence both retinal integrity and skeletal health, rather than isolated organ-specific pathology.

In clinical practice, DEXA is typically reserved for individuals already considered at elevated risk of osteoporosis and is not routinely used as a population-level screening tool due to cost, accessibility, and resource constraints. As a result, osteoporosis screening in primary care often relies on simple risk calculator such as the Osteoporosis Self-assessment Tool (OST), which incorporates calendar age, sex, and body weight. [28] Consistent with this framework, in the cross-sectional PIONEER analyses, higher age, female sex, and lower body weight were the principal non-retinal factors associated with osteoporosis (Table 3), underscoring the robustness of OST-based risk stratification. In the prospective UK Biobank analyses, traditional metabolic comorbidities such as diabetes and hypertension were not significantly associated with incident osteoporosis in the multivariable Cox Ph model. Lifestyle factors, including current smoking and lower physical activity, were associated with osteoporosis risk, but these associations were weaker than those observed for demographic factors and retinal biological age. These findings motivated evaluation of whether retinal biological age could provide incremental prognostic value beyond established OST variables. Accordingly, inclusion of RetiAGE significantly improved longitudinal model fit in likelihood ratio testing, indicating enhanced identification of individuals at elevated fracture risk (S2 Table). Taken together, these findings suggest that retinal biological age may capture information related to skeletal aging that is not fully reflected by conventional demographic or lifestyle risk factors. While DEXA remains the standard for diagnosing osteoporosis, retinal biological age may serve as a complementary marker in research or exploratory screening contexts. Given that retinal aging can precede overt ocular disease, retinal imaging may offer a window into systemic aging processes relevant to bone health.

This study has several limitations. First, the analyses focus on the downstream association and prognostic relevance of a pre-trained retinal biological aging biomarker rather than on real-world model deployment or clinical decision-making. RetiAGE is a probabilistic retinal aging score, for which no direct ground-truth measure of biological age exists. As

RetiAGE was originally developed in a Korean population and applied without population-specific re-training, differences in imaging devices, acquisition protocols, and population characteristics across cohorts may lead to shifts in absolute score distributions. Such heterogeneity is expected to primarily affect calibration rather than relative associations within cohorts and may therefore bias effect the estimates toward the null. Second, although we observed consistent cross-sectional and prospective associations between retinal biological aging, bone mineral density, and osteoporosis-related outcomes across populations differing in ethnicity and calendar age, retinal pigmentation, structure, and vascular patterns may vary by ethnicity. While FRAX-derived fracture risk scores are calibrated using population-specific data, retinal aging models trained or fine-tuned in specific ethnic populations may further optimize performance when applied to ethnicity-specific risk estimation. Finally, the present study focused on a single retinal aging biomarker (RetiAGE). Other retinal aging metrics, such as the retinal age gap (RAG), were not evaluated and may capture complementary aspects of retinal aging biology.

In conclusion, this study examined the association between retinal biological aging and low BMD in an Asian population and further explored the relationship between retinal aging and the onset of osteoporosis in longitudinal data from a European population. These findings highlight a non-invasive and repeatable method for screening low BMD and stratifying the risk of osteoporosis.

## Supporting information

**S1 Table. Variables field IDs and coding extracted from the UK Biobank.**
(DOCX)

**S2 Table. Performance for predicting osteoporotic fracture risk using the Osteoporosis Self-assessment Tool (OST) with and without RetiAGE enhancement.**
(DOCX)

**S3 Table. Association between RetiAGE and risk of osteoporosis during follow-up after adjustment for age and gender.**
(DOCX)

**S4 Table. Sensitivity analysis of the association between RetiAGE and osteoporosis in participants without major ocular diseases affecting fundus appearance in the cross-sectional PIONEER cohort.**
(DOCX)

**S5 Table. Sensitivity analysis of the association between RetiAGE and risk of osteoporosis in White participants in the prospective UK Biobank cohort.**
(DOCX)

**S6 Table. Sensitivity analysis of the association between RetiAGE and risk of osteoporosis in participants without ocular conditions potentially affecting retinal appearance in the prospective UK Biobank cohort.**
(DOCX)

**S7 Table. Sensitivity analysis of the association between RetiAGE and risk of osteoporosis in participants without cataract surgery in the prospective UK Biobank cohort.**
(DOCX)

**S8 Table. Sensitivity analysis of the association between RetiAGE and risk of osteoporosis in participants aged over 60 years in the prospective UK Biobank cohort.**
(DOCX)

**S9 Table. Summary of the genome-wide analysis of the RetiAGE score and the implicated genes.**
(DOCX)

**S10 Table. Distribution of eye diseases and conditions that may affect retinal imaging.**
(DOCX)

**S11 Table. Associations between osteoporosis risk factors and RetiAGE z-score in the cross-sectional PIONEER study.**
(DOCX)

**S12 Table. Associations between osteoporosis risk factors and RetiAGE z-score in the prospective UK Biobank cohort.**
(DOCX)

**S13 Table. Characteristics of the development set and internal test set of the Korean Health Screening Study.**
(DOCX)

**S1 Fig. Study population flowchart.**
(DOCX)

**S2 Fig. Association of the retinal biological age marker with bone mineral density and T-score in univariate linear models.**
(DOCX)

**S3 Fig. Net reclassification improvement estimation for osteoporosis risk at 10 years of follow-up.**
(DOCX)

**S4 Fig. Distribution of RetiAGE scores across different ethnicities (White, Chinese, Indian, and Malay).**
(DOCX)

## Author contributions

**Conceptualization:** Qingsheng Peng, Ching-Yu Cheng.

**Data curation:** Preeti Gupta, Ah Young Leem.

**Formal analysis:** Qingsheng Peng, Kenon Chua, Hengtong Li.

**Funding acquisition:** Ching-Yu Cheng.

**Investigation:** Qingsheng Peng.

**Methodology:** Qingsheng Peng, Kenon Chua, Hengtong Li, Marco C.Y. Yu.

**Project administration:** Qingsheng Peng, Ching-Yu Cheng.

**Supervision:** Ecosse Lamoureux, Ching-Yu Cheng.

**Validation:** Marco C.Y. Yu.

**Visualization:** Qingsheng Peng.

**Writing – original draft:** Qingsheng Peng, Can Can Xue.

**Writing – review & editing:** Qingsheng Peng, Can Can Xue, Kenon Chua, Simon Nusinovici, Zhi Da Soh, Preeti Gupta, Ah Young Leem, Marco C.Y. Yu, Charumathi Sabanayagam, Tyler Hyungtaek Rim, Zhuoting Zhu, Ecosse Lamoureux, Ching-Yu Cheng.

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
