## [Decision Letter · Decision Letter 0]

3 Nov 2025

Response to Reviewers'. This file does not need to include responses to any formatting updates and technical items listed in the 'Journal Requirements' section below.'. This file does not need to include responses to any formatting updates and technical items listed in the 'Journal Requirements' section below.* A marked-up copy of your manuscript that highlights changes made to the original version. You should upload this as a separate file labeled 'Revised Manuscript with Track Changes'.'.* An unmarked version of your revised paper without tracked changes. You should upload this as a separate file labeled 'Manuscript'.'. If you would like to make changes to your financial disclosure, competing interests statement, or data availability statement, please make these updates within the submission form at the time of resubmission. Guidelines for resubmitting your figure files are available below the reviewer comments at the end of this letter. We look forward to receiving your revised manuscript. Kind regards, Pengxu WeiAcademic EditorPLOS Digital Health Pengxu WeiAcademic EditorPLOS Digital Health Leo Anthony CeliEditor-in-ChiefPLOS Digital Healthorcid.org/0000-0001-6712-6626 **Journal Requirements:** If the reviewer comments include a recommendation to cite specific previously published works, please review and evaluate these publications to determine whether they are relevant and should be cited. There is no requirement to cite these works unless the editor has indicated otherwise.  **Additional Editor Comments (if provided):** The cost of retinal photographs is comparable to that of dual-energy X-ray absorptiometry

(DXA). Moreover, DXA involves only minimal radiation exposure and remains the gold standard for

assessing bone mineral density (BMD) and diagnosing osteoporosis. Therefore, if retinal

photographs merely confirm the association that “accelerated retinal biological aging is

associated with decreased BMD and an increased risk of osteoporosis and related fractures,”

this finding alone is not sufficiently compelling.

However, if retinal photographs—particularly when combined with AI-driven analysis—can yield

information not detectable by DXA alone, this would represent a meaningful and potentially

valuable advance. For instance, consider Table 3 (which shows that retinal age predicts

osteoporosis and the risk of osteoporotic fracture even after adjusting for established risk

factors), a critical question then arises: does retinal age offer predictive value for fracture

risk beyond that provided by conventional osteoporosis diagnosis based on DXA?

Specifically, do models using DXA-defined osteoporosis and models using retinal age differ in

their ability to predict fractures? If so, can retinal age provide better prognostic

information? And, most importantly, does this additional information translate into tangible

clinical utility—such as improved risk stratification, earlier intervention, or personalized

prevention strategies? Addressing these questions is essential to determine whether retinal

imaging coupled with AI represents a clinically relevant tool in osteoporosis and fracture risk

**Reviewers' Comments:** Reviewer's Responses to Questions

**Comments to the Author**

1. Does this manuscript meet PLOS Digital Health’s publication criteria? Is the manuscript technically sound, and do the data support the conclusions? The manuscript must describe methodologically and ethically rigorous research with conclusions that are appropriately drawn based on the data presented.? Is the manuscript technically sound, and do the data support the conclusions? The manuscript must describe methodologically and ethically rigorous research with conclusions that are appropriately drawn based on the data presented.

Reviewer #1: Partly

Reviewer #2: No

Reviewer #3: Partly

2. Has the statistical analysis been performed appropriately and rigorously?

Reviewer #1: Yes

Reviewer #2: Yes

Reviewer #3: No

3. Have the authors made all data underlying the findings in their manuscript fully available (please refer to the Data Availability Statement at the start of the manuscript PDF file)?

The PLOS Data policy requires authors to make all data underlying the findings described in their manuscript fully available without restriction, with rare exception. The data should be provided as part of the manuscript or its supporting information, or deposited to a public repository. For example, in addition to summary statistics, the data points behind means, medians and variance measures should be available. If there are restrictions on publicly sharing data—e.g. participant privacy or use of data from a third party—those must be specified.requires authors to make all data underlying the findings described in their manuscript fully available without restriction, with rare exception. The data should be provided as part of the manuscript or its supporting information, or deposited to a public repository. For example, in addition to summary statistics, the data points behind means, medians and variance measures should be available. If there are restrictions on publicly sharing data—e.g. participant privacy or use of data from a third party—those must be specified.

Reviewer #1: Yes

Reviewer #2: No

Reviewer #3: No

4. Is the manuscript presented in an intelligible fashion and written in standard English?

Reviewer #1: Yes

Reviewer #2: Yes

Reviewer #3: Yes

Reviewer #1: Dear Editor and Authors,

I thank the Authors for their submission and for the work invested in this study. A very interesting idea for a big unmet need. Kudos to the team.

General comments:

Abstract:

This is clear and well-structured, but the use of mechanistic phrasing (e.g., positioning retinal aging as an “indicator” of osteoporosis risk) may overstate causality. I suggest softening this language to emphasise that retinal aging is associated with (rather than definitively indicative of) bone outcomes.

Introduction:

The Introduction provides a good overview of retinal biomarkers and bone health. However, it would benefit from a more balanced framing of retinal age as an emerging biomarker, including a brief acknowledgment that prior evidence linking retinal aging to systemic outcomes (including BMD) is preliminary and largely exploratory.

Rows 61–66:

The IRF4 pathway is indeed relevant to bone remodeling and osteoporosis (Wu et al., 2024; Weivoda & Bradley, 2023), and your group has made important contributions to retinal biological age as a biomarker (Nusinovici et al., 2022; Zhu et al., 2022). At present, however, large GWAS of retinal age gap and related traits highlight loci such as ALKAL2, OCA2/HERC2, and POC5 rather than IRF4 (Ahadi et al., 2023; Goallec et al., 2021). This suggests that while the IRF4–bone connection is biologically established, its role in retinal aging is still more inferential. The association between retinal aging and bone outcomes, including osteoporosis, is promising but remains preliminary and mostly reported at an abstract level (Peng et al., 2025, ARVO abstract). To strengthen the rationale, I suggest either (i) reframing this section as a hypothesis or potential biological bridge, or (ii) if the authors believe the IRF4–retina–bone pathway has stronger validation, providing additional evidence or references to support it.

Methods:

The description of the statistical models is relatively brief. Could the authors clarify (i) whether any corrections for multiple comparisons were applied given the number of outcomes tested, (ii) whether interaction terms (e.g., age, sex) were evaluated, and (iii) how missing data were handled in covariates?

Rows 97–99:

The methods section states that the “macula-centered retinal photograph with the best quality judging by an in-house quality control model” was chosen for each eye, and that scores were averaged across two eyes. This is somewhat vague. Could the authors provide more details on: (i) the criteria or thresholds used by the quality control model, (ii) whether images from both eyes were always available and included, and (iii) how missing or poor-quality images were handled in the averaging process?

Rows 102–103:

The description of the DEXA scans as using “very low dose X-rays (~0.005 to 0.01 mSv)” could benefit from clarification. Was this range measured in your study, taken from manufacturer specifications, or cited from prior literature? It may be helpful to specify how the dose was determined, and to explain what “very low dose” means (e.g., compared to background radiation or a chest X-ray). Also, the placement of the reference (20) should be integrated into the sentence (currently appears as a typo). The same formatting issue appears elsewhere in the manuscript and may need systematic correction.

Rows 105–106:

The description of T-scores in the Methods is a bit unclear. It currently states that the T-score compares each participant’s BMD to “the peak bone mass of a healthy 30-year-old adult.” Could the authors clarify how this reference value is determined? Is it based on a fixed age (30 years), or on a manufacturer-provided young adult reference database built into the DEXA system?

Rows 113–120:

Given that retinal structure, pigmentation, and vascular patterns can vary across ethnic groups, and that your algorithm was originally developed in a Korean population, it would be helpful if the manuscript could clarify how well the RetiAGE model performs across the different ethnicities represented in PIONEER (Chinese, Indian, Malay) and in the predominantly European UK Biobank cohort. Since you also use FRAX scores, which are themselves calibrated to population-specific fracture and mortality data, it might be worth commenting on how ethnicity could influence both retinal age predictions and the fracture risk assessment.

Results:

The Results present associations between RetiAGE and bone outcomes, but the magnitude of effect sizes is not always discussed in terms of clinical relevance. For example, how do the observed differences in BMD or fracture risk compare to known risk factors such as smoking, BMI, or diabetes?

Figures and Tables:

Figures are generally clear, though the legends could be more descriptive. For example, specifying the covariates adjusted for in regression models directly in the figure legends would make them easier to interpret without needing to cross-reference the Methods.

Discussion:

One area that may warrant further clarification is the role of ethnicity. In the Methods, it would indeed be useful to explain how the RetiAGE algorithm, originally developed in a Korean population, was applied to the PIONEER cohort (Chinese, Indian, Malay) and the predominantly European UK Biobank, and whether any performance differences were observed across groups. Here, in the Discussion, a brief acknowledgment that retinal pigmentation and vascular patterns may vary by ethnicity, and that FRAX scores are themselves calibrated to population-specific fracture and mortality risks, would strengthen this manuscript.

It might be worth briefly mentioning “non-IRF4” pathways that could explain shared biology between retinal and bone aging, such as systemic inflammation, microvascular changes, or oxidative stress.

Reviewer #2: Review for “Retinal Biological Age Correlates with Bone Mineral Density and Fracture Risk Score and Predicts Incident Osteoporosis”

Interesting and clinically relevant question linking retinal biological aging to bone health and future osteoporosis. However, the manuscript would benefit from clearer methods, harmonized reporting across datasets, and editorial fixes.

Comments

* The manuscript states that RetiAGE reflects the likelihood of being ≥65 years. Please justify the choice of this threshold. Is ≥65 a clinically motivated cut-off, or an artifact of the training data distribution?

* Provide the age distribution of the training population and the disease distribution (including relevant ophthalmic conditions) to contextualize the biomarker.

* Even if previously published, please summarize key model details for reproducibility: deep learning architecture, input size/preprocessing, data augmentation, loss/optimizer, training/validation splits, calibration procedure, and any external validation.

* Given that PIONEER includes individuals >60 years, discuss whether ≥65 remains an optimal threshold and how this affects discrimination/calibration in PIONEER and UK Biobank.

* The current methods interleave two datasets in a way that is hard to follow. I suggest restructuring into parallel subsections for PIONEER and UK Biobank, each with: participants, imaging protocol, RetiAGE inference, outcomes/definitions, covariates, exclusions, and statistical analysis.

* When discussing generalizability, explicitly state whether the two cohorts (RetiAGE training cohort vs PIONEER/UKB) are comparable in demographics and imaging protocols, and whether any harmonization or adjustment was performed.

* Specify the quality-control criteria for fundus images (algorithm, threshold, and performance metrics). Report the proportion excluded and whether excluded participants differ on key variables (age, BMD, comorbidities).

* Clarify comparability between the RetiAGE training cohort and PIONEER: are distributions of age, sex, ethnicity, camera type, and dilation similar? Include a short generalizability discussion or sensitivity analyses if there are domain shifts (e.g., different devices/protocols).

* Please clarify the ophthalmologic disease distribution in both datasets (e.g., diabetic retinopathy, AMD), as these may influence retinal aging estimates.

* You use the average of two eyes. Please report the agreement between eyes and, if possible, provide a worst-eye or per-eye sensitivity analysis.

* Report the prevalence of retinal abnormalities in PIONEER and any exclusion criteria related to ocular disease that could bias RetiAGE.

* Provide the distribution of osteoporosis specifically in participants ≥65 years, to align with the RetiAGE threshold.

* Consider adding descriptive results for RetiAGE distributions (e.g., mean/SD, quartiles) in each cohort and simple associations with key covariates, to help readers interpret its behavior.

Editorial and consistency issues

* Abstract (lines 12–13): typo “uisng”

* Line 240: “dataset” is strikethrough

* Line 319: typo “propective”

* Supplementary Table: fix “Caucassian” → “White” (and I recommend avoiding the term “Caucasian”). The definition “British, Irish, White, or any other white background” does not correspond to the anthropologic term “Caucasian”; use “White” (as per dataset coding) or the explicit categories. Generalizing the entire British and Irish population as white is not correct.

* References: the citation style shifts around references 3–4.

Reviewer #3: The paper is about a novel association between retinal biological age and BMD & incident osteoporosis across cohorts.

Although this is a secondary implementation of the RetiAGE model, it is still a prediction task where RetiAGE values are used as a predictor for BMD & incident osteoporosis, and thus these warrant detailing of the model's features as per the TRIPOD framework. Below are certain recommendations that will improve the paper significantly:

1. No mention of Outliers: There is no mention of removal of outliers or the methods used for identification and removal.

2. No internal/external validation of an osteoporosis prediction model.

3. No Github code/ supplementary material required for reproducibility.

4. PIONEER dataset is not publicly available.

5. No discrimination/calibration/ interpretability metrics used.

Overall, the paper shows promising cross-cohort associations, but transparency, TRIPOD reporting, predictive performance, calibration, and reproducibility need substantial strengthening.

**Do you want your identity to be public for this peer review?** For information about this choice, including consent withdrawal, please see our Privacy Policy..

Reviewer #1: No

Reviewer #2: No

Reviewer #3: No

**Figure resubmission:**  While revising your submission, we strongly recommend that you use PLOS’s NAAS tool (https://ngplosjournals.pagemajik.ai/artanalysis) to test your figure files. NAAS can convert your figure files to the TIFF file type and meet basic requirements (such as print size, resolution), or provide you with a report on issues that do not meet our requirements and that NAAS cannot fix. 

**Reproducibility:** To enhance the reproducibility of your results, we recommend that authors of applicable studies deposit laboratory protocols in protocols.io, where a protocol can be assigned its own identifier (DOI) such that it can be cited independently in the future. Additionally, PLOS ONE offers an option to publish peer-reviewed clinical study protocols. Read more information on sharing protocols at https://plos.org/protocols?utm_medium=editorial-email&utm_source=authorletters&utm_campaign=protocols  To enhance the reproducibility of your results, we recommend that authors of applicable studies deposit laboratory protocols in protocols.io, where a protocol can be assigned its own identifier (DOI) such that it can be cited independently in the future. Additionally, PLOS ONE offers an option to publish peer-reviewed clinical study protocols. Read more information on sharing protocols at https://plos.org/protocols?utm_medium=editorial-email&utm_source=authorletters&utm_campaign=protocols 

---

## [Decision Letter · Decision Letter 1]

29 Mar 2026

Retinal Biological Age Correlates with Bone Mineral Density and Fracture Risk Score and Predicts Incident Osteoporosis

PDIG-D-25-00623R1

Dear Dr. Cheng,

We are pleased to inform you that your manuscript 'Retinal Biological Age Correlates with Bone Mineral Density and Fracture Risk Score and Predicts Incident Osteoporosis' has been provisionally accepted for publication in PLOS Digital Health.

Best regards,

Pengxu Wei

Academic Editor

PLOS Digital Health